# A Triazaspirane Derivative Inhibits Migration and Invasion in PC3 Prostate Cancer Cells

**DOI:** 10.3390/molecules28114524

**Published:** 2023-06-02

**Authors:** Javier de Jesús Vasconcelos-Ulloa, Victor García-González, Benjamín Valdez-Salas, José Gustavo Vázquez-Jiménez, Ignacio Rivero-Espejel, Raúl Díaz-Molina, Octavio Galindo-Hernández

**Affiliations:** 1Instituto de Ingeniería, Universidad Autónoma de Baja California, Mexicali 21100, Baja California, Mexico; javier.vasconcelos@uabc.edu.mx (J.d.J.V.-U.); benval@uabc.edu.mx (B.V.-S.); 2Facultad de Medicina Mexicali, Universidad Autónoma de Baja California, Mexicali 21000, Baja California, Mexico; vgarcia62@uabc.edu.mx (V.G.-G.); gustavo.vazquez@uabc.edu.mx (J.G.V.-J.); 3Laboratorio Multidisciplinario de Estudios Metabólicos y Cáncer, Universidad Autónoma de Baja California, Mexicali 21000, Baja California, Mexico; 4Centro de Graduados e Investigación en Química, Instituto Tecnológico de Tijuana, Tijuana 22000, Baja California, Mexico; irivero@tectijuana.mx

**Keywords:** prostate cancer, PC3, triazaspirane, FAK, Src

## Abstract

Cancer is a serious health problem due to the complexity of establishing an effective treatment. The purpose of this work was to evaluate the activity of a triazaspirane as a migration and invasion inhibitor in PC3 prostatic tumor cells through a possible negative regulation of the FAK/Src signal transduction pathway and decreased secretion of metalloproteinases 2 and 9. Molecular docking analysis was performed using Moe 2008.10 software. Migration (wound-healing assay) and invasion (Boyden chamber assay) assays were performed. In addition, the Western blot technique was used to quantify protein expression, and the zymography technique was used to observe the secretion of metalloproteinases. Molecular docking showed interactions in regions of interest of the FAK and Src proteins. Moreover, the biological activity assays demonstrated an inhibitory effect on cell migration and invasion, an important suppression of metalloproteinase secretion, and a decrease in the expression of p-FAK and p-Src proteins in treated PC3 cells. Triazaspirane-type molecules have important inhibitory effects on the mechanisms associated with metastasis in PC3 tumor cells.

## 1. Introduction

Prostate cancer (PCa) is a global public health problem. In 2020, PCa had the fourth highest incidence, accounting for 7.3% of all cancers (1,414,259 reported cases) and the fifth highest mortality rate, accounting for 3.8% (375,304 deaths) of all cancers [1]. Cancer is considered a heterogeneous disease and is highly variable due to the constant change in its molecular characteristics. 

Metastasis is a biological process in which tumor cells spread to other organs, with the main target organs being bone, lungs, liver, and the brain. In this phenomenon, tumor cells acquire the ability to migrate and invade nearby tissues, such as pelvic lymph nodes, and eventually spread to other organs. Metastasis is the leading cause of cancer death, resulting in up to 90% of deaths in patients with cancer [2]. PCa is usually treated using local surveillance and radiotherapy or radical prostatectomy; treatment choice is based on clinical status and prostate-specific antigen levels. When the disease is in advanced stages and has already invaded other tissues, androgen deprivation therapy by surgical or chemical castration becomes an option; however, relapses can occur within 18–36 months, suggesting resistance to treatment [3]. During prostate cancer treatment, patients may develop urinary, sexual, or intestinal dysfunction, resulting in poor improvement in their quality of life [4]. Therefore, pharmacological alternatives should be developed to improve outcomes in patients with PCa. Azaspiran-type molecules are compounds that have been associated with various beneficial effects, including their immunoregulatory capacity in murine models with rheumatoid arthritis, decreasing the inflammatory response and resulting in an improvement in the preservation of bone integrity [5,6]. 

On the other hand, antitumor effects have been described in azaspiran-type molecules. In myeloma cells, N′N-diethyl-8,8-dipropyl-2-azaspiro[4,5]decane-2-propanamine (atiprimod) exhibits antiproliferative and antiangiogenic properties. Atiprimod modulates these processes through activation of proapoptotic proteins such as caspase-3 and -8, inhibition of Akt and STAT3 phosphorylation, and decreased production of vascular endothelial growth factor (VEGF) and interleukin. 6 (IL-6) [7,8], coupled with a decrease in the protein expression levels of JAK2 and p-JAK2, negatively regulating the JAK–STAT pathway [9,10]. In line with this notion, the antiproliferative effect of a derivative of the type 1 oxa-3-azaspiro[5.5]undecane has been observed through the modulation of the JAK2/STAT3 pathway in hepatocellular carcinoma and breast cancer cells, regardless of estrogen receptor expression status [11,12]. Various studies have proposed new strategies for the improvement of the quality of life of patients with PCa. Sulaiman et al. [11] demonstrated that 2-(1-(4-(2-cyanophenyl)1-benzyl-1H-indol-3-yl)-5-(4-methoxy-phenyl)-1-oxa-3-azaspiro(5,5)undecane (CIMO) suppresses colony formation and viability of MDA-MB-231 and MCF-7 breast cancer cells by inhibiting the overactivation of the STAT3 pathway, which is strongly related to the progression of different types of tumors. STAT3 has been associated with cancer cell growth, particularly in hepatocellular carcinoma, lymphoma, and multiple myeloma, in the Appendix A. Some azaspiran-type molecules (Atiprimod and CIMO) are observed, in addition to the structure of the molecule TRI-BE. The aim of the present study was to evaluate the inhibitory effect of 8-benzyl-1,3,8-triazaspiro[4.5]decane-2,4-dione (TRI-BE) on PC3 prostate cancer cells of a highly invasive and metastatic cell line, as well as migration and invasion.

This is an explanatory, experimental, quantitative, in silico, and in vitro study using the PC3 cell line. Our in silico data suggest that TRI-BE can interact with FAK and Src. Accordingly, TRI-BE induces a decrease in FAK and Src phosphorylation levels, accompanied by a decrease in MMP-9 secretion in PC3 cells. However, TRI-BE does not cause statistically significant changes in E-cadherin and N-cadherin expression. Finally, TRI-BE induces a decrease in tumor migration and invasion. In summary, our data show, for the first time, that TRI-BE can inhibit PC3 prostate tumor cell migration and invasion.

## 2. Results

### 2.1. Molecular Docking: TRI-BE vs. FAK

In the present work, we proposed to evaluate various phenomena associated with tumor progression, including cell migration and invasion, that are significantly related to the FAK/Src signaling pathway. Therefore, a series of molecular docking tests was performed between TRI-BE (the macromolecule) and FAK (the protein). The FAK structure was obtained from the Protein Data Bank: 1MP8 crystal structure of focal adhesion kinase (FAK) (PDB https://doi.org/10.2210/pdb1MP8/pdb). The results are shown in Table 1; the three highest-affinity interactions between the conformers obtained from triazaspirane with FAK can be observed. The highest-affinity value obtained an E-score of −7.7878 kcal/mol, the interaction of which is shown in Figure 1A; followed by the conformer, which exhibited an interaction with an E-score of −7.7637 kcal/mol (Figure 1B); and finally, the interaction with an E-score of −7.6901 kcal/mol corresponding to the interaction of a third conformer (Figure 1C).

The most representative interactions of the different conformers obtained from the docking between TRI-BE and FAK (Figure 1) were as follows: a cation–pi interaction between the TRI-BE and Arg 598 residue and a hydrogen bond interaction between the TRI-BE and Ser 664 (Figure 1A); a hydrogen bond interaction between the TRI-BE and the Thr 491 residue side chain (Figure 1B); and a hydrogen bond interaction between the TRI-BE and the Thr 600 side chain (Figure 1C).

### 2.2. Molecular Docking: TRI-BE vs. Src

To study the possible interaction between TRI-BE and Src using molecular docking, the Src structure was obtained from the PDB database. The 1FMK crystal structure of human tyrosine-protein kinase C-Src (PDB https://doi.org/10.2210/pdb1FMK/pdb) was selected as the protein. Table 2 shows the three most significant E-score results. Figure 2A shows the highest-affinity interaction (E-score: −8.2888 kcal/mol), Figure 2B shows the average E-score interaction (−7.9860 kcal/mol), and Figure 2C shows the lowest-affinity interaction (E-score: −7.8069 kcal/mol).

The most representative interactions of the different conformers of TRI-BE and Src (Figure 2) were as follows: a hydrogen bond interaction between TRI-BE and residue Glu 147 residue and another hydrogen bond interaction with Tyr 90 residue side chain (Figure 2A); no direct interactions between TRI-BE and Src (Figure 2B); and a hydrogen bond interaction with the Glu 147 residue (Figure 2C).

The three conformers with the lowest binding energy were selected, as they are indicative of a higher degree of affinity to bind to the amino acids of a specific region of the FAK and Src proteins. These values are influenced by the interactions present in the coupling, such is the case of van der Waals, electrostatic, hydrogen bond, and hydrophobic interactions [13]. As shown by the interactions presented in Figure 3A, the coupling of TRI-BE with FAK, at least theoretically, has a great affinity for the region of the kinase domain, which is an important domain, since it generates the phosphorylation of target proteins downstream of FAK that lead to adhesion, spreading, migration, and metastasis. Moreover, catalytic site-binding proteins such as FIP200 have been shown to inhibit FAK activity [14].

On the other hand, in combination with the Src protein shown in Figure 3B, the selected TRI-BE conformers show higher affinity for the SH2 and SH3 domains, which are for interaction with proteins. In particular, the binding of TRI-BE to the SH2 domains in Src would block the recognition and interaction with the phosphorylated Y397 residue of FAK, suppressing the activation of the FAK/Src pathway [15] accompanied by a lower migratory and invasive capacity of cancer cells. This mechanism would explain the inhibitory effect of TRI-BE on the migration and invasion of PC3 cells.

### 2.3. TRI-BE Inhibits PC3 Cell Migration

First, we evaluated the impact of TRI-BE on cell viability. Our data indicate that 10 μM TRI-BE does not affect the viability of PC3 cells. We decided to study the effect of TRI-BE on PC3 cell migration using wound-healing assays. PC3 cells treated with TRI-BE showed a lower migratory capacity than control cells (Figure 4A) and exhibited the greatest inhibitory effect at 50 µM.

### 2.4. TRI-BE Promotes a Decrease in the Invasive Capacity of PC3 Tumor Cells

The negative effect of TRI-BE on cell invasion was observed over a 24 h time course of stimulation (Figure 5A), suggesting that cell invasiveness was significantly inhibited upon application of TRI-BE compared to the positive control with 5% fetal bovine serum.

### 2.5. TRI-BE Induces a Decrease in MMP-9 Secretion

TRI-BE at a concentration of 10 µM showed a remarkable suppressive effect on MMP-9 secretion (Figure 6A,B). TRI-BE reduced MMP-9 secretion by more than 50% (Figure 6B).

### 2.6. TRI-BE Promotes a Decrease in Phosphorylation Levels of FAK and Src

We evaluated the effect of TRI-BE in regulating the activity of the FAK/Src signaling pathway and its impact on cell migration. As seen in Figure 7A, TRI-BE treatment induced a decrease in phosphorylated FAK (p-FAK) levels, with maximum inhibition at 5 min. Inhibitory effects were observed at the three time courses; however, TRI-BE had the greatest effect after 30 min of exposure.

Figure 8A shows the inhibitory effect of TRI-BE on phosphorylated Src (p-Src) expression, which was highest after 30 min of exposure. The results shown in Figure 8B correspond to three independent experiments.

### 2.7. Evaluation of E- and N-Cadherin Expression

Figure 9A shows that there was no significant change in E-cadherin expression after the addition of TRI-BE after a 24 h time course of exposure. After statistical analysis of three independent studies, very similar values were observed in both evaluated treatments (Figure 9B, with and without TRI-BE and FBS).

TRI-BE tended to suppress N-cadherin expression (Figure 10A); however, no statistically significant changes were evident between the compared treatments (Figure 10B).

## 3. Discussion

Prostate cancer is a major global health problem and is the most prevalent cancer in Mexican men, representing 29.9% of new cases of cancer in men (26,742 cases) in 2020 [1]. This highlights the need for diagnostic and therapeutic alternatives to improve the quality and life expectancy of patients with prostate cancer, especially in hormone-refractory tumors. One of the main problems faced by patients with advanced prostate cancer is resistance to androgen deprivation therapy (ADT), which is performed by surgical or chemical castration. However, even with ADT, the disease will progress to castration-resistant prostate cancer (CRPC). Approximately 90% of patients with CRPC develop bone metastases, which are related to mortality associated with the disease [16,17]. In recent years, a wide variety of molecules with anticancer activity has been developed and tested, particularly the use of spiro-heterocyclic compounds [18]. These molecules have been studied as antitumor agents, with favorable results in different types of cancer, such as myeloma, breast cancer, and acute myeloid leukemia [18,19,20,21]. Considering these findings, we propose determining the potential regulatory effect of triazaspirane derivatives with spiro-hydantoin-based structure (TRI-BE) on the mechanisms of biological activity in cancer cells and elucidating their possible association with the FAK/Src signaling pathway involved in the development of cancer [15,22].

Spiro-type compounds are those that have a structure formed by two orthogonally joined rings; these compounds usually have different heterocyclic components, managing to acquire a certain similarity with some macromolecules, such as proteins, allowing them to interact with proteins in complex biological systems [18].

In multiple cancer models, azaspirans and their derivatives have been tested as inhibitors of some tyrosine kinase proteins related to tumor progression, such as atiprimod (JAK2/JAK3 inhibitor) and lestaurtinib (JAK2 inhibitor), which present different tumor inhibition mechanisms [11,12].

In particular, the cyclic structure of TRI-BE is the spiro-hydantoin type, and hydantoin has been used for a wide variety of pathologies, most notably owing to its anticonvulsant, anti-inflammatory, and antitumor properties, among others [23].

An interesting aspect regarding compounds with potential biological activity is the evaluation of the application of Lipinski’s rules, which empirically determine the potential solubility and permeability of a molecule in a biological system. Analysis of a molecular structure based on these rules is conclusively accepted because they are based on certain molecular properties, such as having an octanol/water partition coefficient (log P) ≤ 5, with a molecular weight ≤ 500 amu, containing no more than five hydrogen bridge donors and no more than ten bond hydrogen bridge acceptors. Lipinski’s rules have a major impact on human pharmacokinetics. TRI-BE meets these aforementioned criteria, suggesting that it can be potentially used as a cancer drug, owing to its ability to diffuse in a biological system [24].

Cell migration and invasion are closely related to tumor development and are crucial factors in metastases, which, in turn, are considered the main cause of death in patients with cancer [25,26]. After identifying migration and invasion as fundamental processes in tumor progression, we focused on the FAK/Src pathway, which is related to cell motility [14]. FAK is of great interest since it has been reported to be overactivated in various tumors, is considered a biomarker for metastasis [27], and is associated with different pathways related to cell survival and invasion [28]. Therefore, FAK and Src were selected for molecular docking studies with TRI-BE. In silico analyses, docking showed various interactions between FAK and TRI-BE, suggesting that TRI-BE has an important affinity for Y397, Y576, and Y577 residues located in the kinase domain (Figure 3A). It is important to note that the kinase domain is highly relevant because it interacts with various stimuli, such as those associated with G proteins. It is also the region where FAK is autophosphorylated, leading to Src recruitment, which is mediated by high affinity to the SH2 region of the Src and results in the phosphorylation of various sites (Y576, Y577) in FAK, allowing the protein to reach its greatest catalytic capacity [29]. Furthermore, in the docking analysis of Src, a considerable affinity to TRI-BE was observed in the FAK-binding region, the SH2 region (Figure 3B). Therefore, we can deduce a possible inhibitory effect exerted on this signaling pathway, which may serve as the basis for subsequent biological activity studies.

The migratory and invasive capacity of tumor cells is a major challenge in the field of oncology. Overactivation of pathways, such as FAK/Src, promote phenomena such as those mediated by GTPases (for example, Cdc42, Rac, and RhoC) and causes the rearrangement of the cellular cytoskeleton through the formation of lamellipodia and the loss of cell–cell interactions mediated by E-cadherin. This promotes the acquisition of an invasive phenotype in the cell through EMT, which represents an important characteristic for metastatic development, a phenomenon that seriously compromises the life expectancy of patients with cancer [30,31]. In the present work, cell migration studies confirmed that TRI-BE significantly inhibits PC3 cell migration. These results are comparable to those reported by Sulaiman et al. [11] and Mohan et al. [12], who found that azaspirane derivative CIMO had an inhibitory effect on hepatocarcinoma and breast cancer cell migration, respectively. In our study, a dose–response curve was constructed, with concentrations ranging from 1 to 50 µM, to evaluate the effect of TRI-BE on cell migration. It should be noted that our results showed a favorable trend from 5 µM, a dose similar to that used in the abovementioned studies [11,12]; however, 10 µM was used for subsequent experiments to achieve clear statistical significance. Likewise, in the cell invasion assays, an indispensable study to evaluate tumor progression, we found that TRI-BE at 10 µM had an inhibitory effect against our control FBS, also showing a relationship with the results reported by Sulaiman et al. [11] and Mohan et al. [12].

Another determining factor of tumor progression analysis is related to a study by Juárez et al. [32], who proposed that overactivation of the FAK/Src pathway leads to greater activation and secretion of MMPs, correlating with an increase in tumor proliferation and invasion. In their work, the authors mentioned that the collagenases MMP-2 and -9 are determining factors in the invasion process due to their ability to degrade the basement membrane of tumor cells, owing to their affinity for collagen types IV, V, VII, X, and XIV; gelatin; elastin; galectin-3; and fibronectin [32,33,34,35]. In the present study, an important regulatory capacity for MMP-9 segregation was demonstrated, with statistically significantly lower levels observed when treated with TRI-BE at 10 µM. This finding is important for EMT management; Niland et al. [36] determined that EMT was mediated by MMP activity due to loss of polarity and cell adhesion. Amit-Vazina et al. [8] used the azaspirane 2-(3-diethylaminopropyl)-8,8-dipropyl-2-azaspiro[4.5]decane-dimaleate (atiprimod) and found that it inhibited NF-kB activity in multiple myeloma cells at 8 µM. These results have important association with our findings because NF-kB activity is related to the overexpression of MMPs [37], which could indicate that TRI-BE has a possible regulatory effect on the FAK/Src signaling pathway; however, additional experiments are required to corroborate this hypothesis.

After confirming that the inhibition of PC3 cancer cell migration and invasion by TRI-BE, protein expression assays were performed to elucidate a possible mechanism that explains this inhibitory effect. The expression of the phosphorylated FAK and Src was evaluated, as they are the active forms associated with tumor progression [22,29]. Several compounds, such as TAE-226, which is associated with apoptosis induction; PF562271; and CEP-37440, have already been shown to inhibit FAK kinase. In addition, inhibitors such as PF-271 and Y15, the mechanism of action of which involves block FAK residue Y397, are considered potential blockers of tumorigenesis in breast cancer [38,39,40,41]. The results of this work indicate a suppressive effect of TRI-BE at 10 µM on the expression of p-FAK in PC3 tumor cells. Azaspirane-like molecules, specifically atiprimod, inhibit myeloma cell proliferation by modulation, which involves blocking the phosphorylation of key proteins, such as STAT3, with a maximum inhibition at 20 µM. Conversely, Choudhari et al. [42] noted a decrease in phosphorylated AKT (pS473-AKT) at concentrations of atiprimod ranging from 0.5 to 5 μM in other signaling pathways that determine tumor development, such as JAK-STAT3 and PI3K/Akt [7,11,42]. Taken together, it is worthwhile to evaluate the effect of TRI-BE and other triazaspiranes on other signaling pathways (for example, JAK-STAT3 and PI3K/Akt) and establish their potential effect as molecules that regulate tumor progression.

Src is a protein whose catalytic activity significantly increases in different types of invasive cancer; therefore, inhibiting its activity is important to decrease the migratory and invasive capacity of cancer cells. Preclinical studies have shown that certain drugs, such as AZD0530 (which has an affinity for the ATP binding site in Src) and vandetanib (vascular endothelial growth factor receptor-2, epidermal growth factor receptor, and RET tyrosine kinases inhibitor) have important effects in reversing the invasive phenotype of breast cancer cells. Because Src participates in migration and motility mechanisms, as well as in the regulation of cell survival and proliferation pathways, it is a therapeutic target in tumor cell regulation [43,44]. The results of the present work show that exposure to TRI-BE at 10 µM caused a significant decrease at different exposure times, with a greater inhibitory effect at the longest exposure time (30 min), inducing an important suppressive effect on Src phosphorylation levels.

As mentioned above, one of the factors triggered by the overactivation of the FAK/Src signaling pathway is the modification of cell–cell interactions mediated by cadherins, which are calcium-dependent transmembrane glycoproteins that are vital to the maintenance and shape of cell and tissue structure [45]. Cadherins have been studied in relation to different types of cancer, such as gastric, prostate, and breast cancers [46]. N-cadherin is associated with tumor progression (increase in cell migration and motility), establishing a relationship with its expression and the invasive or mesenchymal phenotype, to the point of being considered a biomarker of EMT. A significant decrease in E-cadherin expression, which is normally expressed in the epithelial phenotype, may affect metastasis [47,48]. Furthermore, cadherin switching involves decreases and increases in E- and N-cadherins, respectively, a phenomenon that is observed in tumor development. The role of E-cadherin is indispensable because a loss of this protein significantly compromises cell stability [49,50]. In this work, we observed that TRI-BE (10 µM) had a marked tendency to improve E-cadherin expression in the treatment with only FBS (under normal conditions for tumor cell growth). Conversely, N-cadherin expression was not significantly affected; however, TRI-BE tended to suppress N-cadherin expression. New experiments should be conducted with the exposure time to TRI-BE increased to 48 h and its concentration increased using a dose–response curve ranging from 10 to 30 µM.

Consequently, TRI-BE has important inhibitory effects on the biological activity of PC3 prostate tumor cells that were demonstrated in the migration and cell invasion assays, along with a decrease in MMP-9 segregation, which led to a decrease in the progression of metastasis in PC3 tumor cells. We propose a regulatory mechanism that involves the suppression of the FAK/Src signaling pathway, which is related to multiple factors associated with tumor progression, such as migration, invasion, and angiogenesis (Figure 11), verifying that the phosphorylation levels of both proteins of interest, FAK and Src, were lower in their phosphorylated (active) form. This elucidates that a negative regulation of the pathway occurred after the interaction with the triazaspirane-like molecule.

## 4. Materials and Methods

### 4.1. Molecular Docking

Molecular docking analysis was performed using the Molecular Operating Environment (MOE) 2008.10 program (Chemical Computing Group Inc., Sherbrooke St. W, Montreal, QC, Canada). TRI-BE ligand design was performed using the Corina program (Molecular Networks GmbH Computerchemie, Nürnberg Germany). FAK and Src were obtained from the Protein Data Bank (PDB): 1MP8 crystal structure of focal adhesion kinase (FAK) (PDB https://doi.org/10.2210/pdb1MP8/pdb) and 1FMK crystal structure of human tyrosine-protein kinase C-Src (PDB https://doi.org/10.2210/pdb1FMK/pdb). The generation of the respective TRI-BE conformers to be evaluated was performed using Moe 2008.10. The proteins of interest, FAK and Src, were prepared for coupling using Moe 2008.10 by eliminating water molecules, protonating, and selecting the force field. Molecular coupling of TRI-BE with FAK or Src was performed by selecting the three highest-affinity E-Scores. The 2D and 3D images of the interaction site between the involved amino acids and the TRI-BE ligand were generated.

### 4.2. Compound Preparation

The TRI-BE molecule was synthesized at the Graduate and Research Center of the Technological Institute of Tijuana, Baja California, according to the methodology described by Rivero et al. [51]. Once the compound was obtained in its pure state, it was diluted in a 20% (*v*/*v*) ethanol solution (vehicle) to a concentration of 1 μM (stock solution). Subsequently, the stock solution was used to stimulate the PC3 cell cultures at the concentrations indicated in the experiments.

### 4.3. Standardization of PC3 Cell Line Culture

The standardization of the PC3 prostate cancer cell line consisted of incubating the cultures in 100 mm diameter boxes with Dulbecco’s Modified Eagle Medium (DMEM) supplemented with 3.7 g/L sodium bicarbonate, 5% FBS, and antibiotic–antimycotic 100× (penicillin and streptomycin, Gibco, ThermoFisher Scientific, Waltham, MA, USA) in a humid atmosphere with 5% CO_2_ and 95% oxygen at 37 °C.

### 4.4. MTT Assay

Cell viability was assessed by an MTT reduction assay under different conditions for 24 h. First, cells were seeded onto 96-well plates at a 150,000 cells/well density and allowed to grow at 90% of confluence. Next, the culture medium was replaced with an Opti-MEM medium. After 2 h under this condition, cells were treated in several experiments. Later, 30 μL of an MTT 2.1 mg/mL stock solution was added to the culture medium to obtain a final concentration of 0.5 mg/mL. Formazan crystals formed after 4 h of incubation were further dissolved by adding buffer lysis (20% sodium dodecyl sulfate, 50% *N*,*N*-dimethylformamide, pH 4.0) according to a previous report [52]. Finally, optical density was measured at 570 nm using a microplate reader.

### 4.5. Wound-Healing or “Scratch” Migration Test

PC3 cell cultures were stimulated with TRI-BE to assess their inhibitory capacity on cell migration using the wound-healing assay (also called “scratch” assay) as previously reported [53]. A PC3 cell culture with 100% confluence was suppressed with DMEM without FBS for 18 h and treated with mitomycin C for 2 h. After suppression, scratching or “wounding” was performed using the tip of a sterile 200 µL pipette; then, cultures were washed once with phosphate-buffered saline (PBS) 1× to remove suspended cells. The PC3 cells were then stimulated for 24 h with TRI-BE. After stimulation, the cells were fixed with cold methanol and stained with Coomassie Brilliant Blue G-250 in 10% acetic acid and 30% methanol (Coomassie Brilliant Blue G-250, Bio-Rad, Hercules, CA, USA) and washed thrice with PBS 1X. Cell migration within the wound was photographed using a Vista Vision-VWR inverted microscope (Radnor, PA, USA) with a Moticam 5 coupled camera (Hong Kong).

### 4.6. Boyden’s Chamber InvasionAssay

Invasion tests were performed in Boyden chambers using filters with 8 µm diameter pores [53]. PC3 cells were suppressed with FBS for 18 h. Matrigel BD (15 µL) and 45 µL of DMEM were placed in the inserts at the top of the chamber, then placed in the incubator for 30 min until the Matrigel BD and DMEM solidified. Subsequently, cell passaging was performed by placing 100 µL of cells per treatment on the solidified Matrigel in the Boyden chamber. A final volume of 600 µL of the corresponding DMEM, DMEM + FBS, and DMEM + FBS + TRI-BE treatments was placed at the bottom of the chamber. The cells were incubated under these treatment conditions for 24 h at 37 °C, after which the medium was aspirated from the compartments; the cells were fixed with cold methanol for 10 min, and cells that did not migrate were removed from the top of the chamber using a cotton swab. After performing this procedure, the cells at the bottom of the chamber were stained with Coomassie Brilliant Blue G-250 for 10 min. Membranes were washed thrice with double distilled water, and finally, the presence of cell invasion was identified by analyzing the Boyden chamber membrane using a Vista Vision-VWR inverted microscope with a Moticam 5 coupled chamber.

### 4.7. Immunodetection (Western Blotting)

PC3 cell cultures stimulated with TRI-BE were lysed for protein quantification using a BCA Pierce™ Protein Assay Kit (ThermoFisher Scientific™, USA) according to the manufacturer’s instructions. These lysates were electrophoresed on acrylamide gels (30%, 29:1 acrylamide/Bis solution, BIO-RAD) with 10% sodium dodecyl sulfate (SDS)-polyacrylamide gel electrophoresis to separate the proteins, which were transferred to a polyvinylidene fluoride (PVDF) membrane. The PVDF membranes were blocked with 5% milk for 2 h, then incubated overnight with the primary antibody diluted to 1:500 at 4 °C. The following antibodies were used: anti-p-FAK-Tyr397, anti-p-Src-Tyr418, anti-E-cadherin, anti-N-cadherin, anti-β-actin, total anti-FAK, and total anti-Src. After incubation, the membranes were washed thrice for 10 min with PBS 1× with 0.1% Tween, then incubated with species-specific secondary antibody (1:5000 dilution) after peroxidation for 2 h at 37 °C with shaking and washed thrice for 10 min with PBS 1× with 0.1% Tween, producing chemiluminescence. Autoradiographs were scanned, and the bands were quantified using ImageJ 1.52a (Bethesda, MD, USA).

### 4.8. Zymography

Confluent PC3 cell cultures were treated with TRI-BE (20% ethanol was used as the vehicle) and only FBS for the positive control. Conditioned (treated) media were collected and concentrated using Centricon centrifugal filters (Millipore) at 2500 rpm for 2 h. Equal volumes of concentrated conditioned medium were mixed with sample buffer (2.5% SDS, 1% sucrose, and 4 µg/mL phenol red) without reducing agents and loaded onto 8% acrylamide gels (30%, 29:1 acrylamide/Bis solution, BIO-RAD) copolymerized with gelatin at 1 mg/mL. The gels were washed twice in 2.5% Triton X-100 and incubated in activation buffer (50 mM Tris-HCl, 5 mM CaCl_2_, pH 7.4) at 37 °C for 24 h. The gels were fixed and stained with Coomassie Brilliant Blue G-250. Proteolytic activity was detected with the presence of clear bands against the stained background of an undigested substrate. Gels were scanned, and bands were quantified using ImageJ 1.52a.

### 4.9. Statistical Analysis

Data were expressed as the mean of each experimental treatment ± standard deviation and subjected to one-way ANOVA with Dunnett’s test. For differences between groups, a significance value (α) of *p* < 0.05 was used. All statistical analyses were performed using GraphPad Prism version 8.0.2 for Windows (Boston, MA, USA) [54].

## 5. Conclusions

The molecule 8-benzyl-1,3,8-triazaspiro[4.5]decane-2,4-dione (TRI-BE) exhibited an inhibitory effect on PC3 prostate tumor cell migration and invasion. Furthermore, MMP-9 segregation decreased in PC3 cells. In addition to a suppression of the expression of phosphorylated FAK and Src after treatment with TRI-BE, which is related to the inhibitory effect of the molecule. This work provides a foundation for future studies on the use of triazaspiranes as regulators of “cadherin switching”.

## Figures and Tables

**Figure 1 molecules-28-04524-f001:**
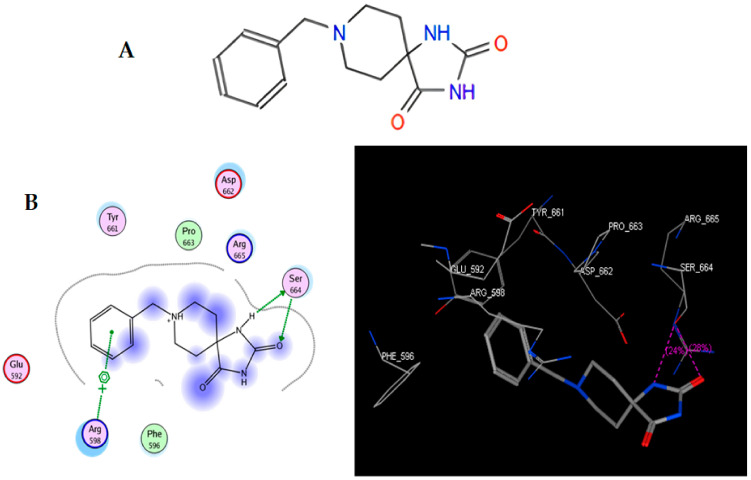
Structure of TRI-BE and molecular docking of TRI-BE and FAK. (**A**) Structure of TRI-BE in 2D. (**B**) The highest-affinity interaction (E-score: −7.7878 kcal/mol). (**C**) Mean affinity interaction (E-score: −7.7637 kcal/mol). (**D**) The lowest-affinity interaction (E-score: −7.6901 kcal/mol). The 2D and 3D structures are shown on the left and right, respectively.

**Figure 2 molecules-28-04524-f002:**
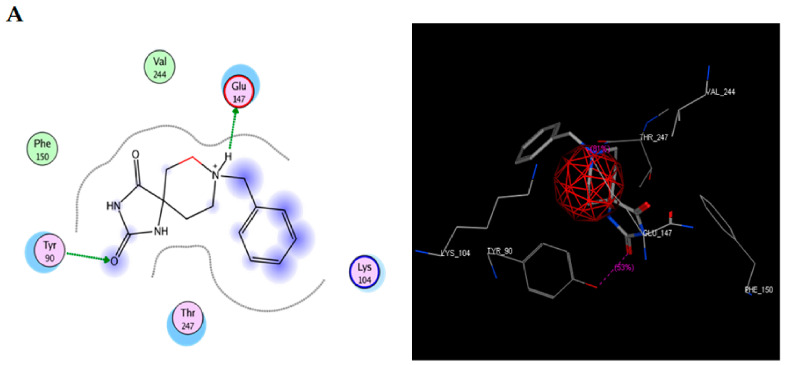
Molecular docking of TRI-BE and Src. (**A**) The highest-affinity interaction (E-score: −8.2888 kcal/mol). (**B**) Mean affinity interaction (E-score: −7.9860 kcal/mol). (**C**) The lowest-affinity interaction (E-score: −7.8069 kcal/mol). The 2D and 3D structures are shown on the left and right, respectively.

**Figure 3 molecules-28-04524-f003:**
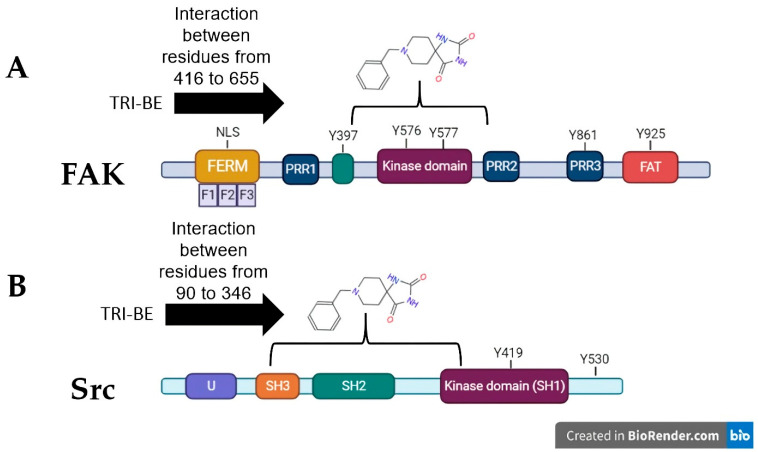
Theoretical interactions between TRI-BE and FAK (**A**) and Src (**B**) evaluated using molecular docking.

**Figure 4 molecules-28-04524-f004:**
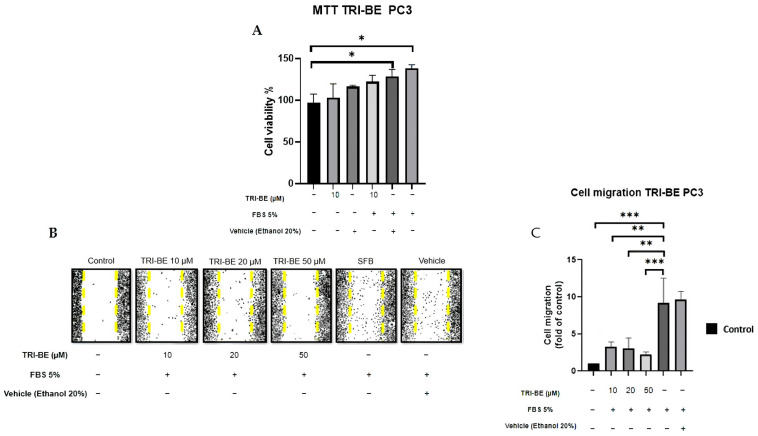
TRI-BE inhibits PC3 cell migration. (**A**) Cell viability was evaluated by MTT assay. (**B**) A 24 h time course. Treatments are plotted from left to right: (control) DMEM; (TRI-BE 10 µM) 10 µM TRI-BE + DMEM + FBS 5%; (TRI-BE 20 µM) 20 µM TRI-BE + DMEM + FBS 5%; (TRI-BE 50 µM) 50 µM TRI-BE + DMEM + FBS 5%; (FBS) DMEM + FBS 5%; (vehicle) DMEM + ethanol 20%. (**C**) The graph represents the means of each treatment ± SD of at least three independent experiments. CTRL = control, DMEM = Dulbecco’s Modified Eagle Medium, FBS = fetal bovine serum, SD = standard deviation. Asterisks indicate statistical significance; * *p* < 0.05, ** *p*< 0.01, *** *p* < 0.001, as assessed using one-way analysis of variance (ANOVA) with Dunnett’s test.

**Figure 5 molecules-28-04524-f005:**
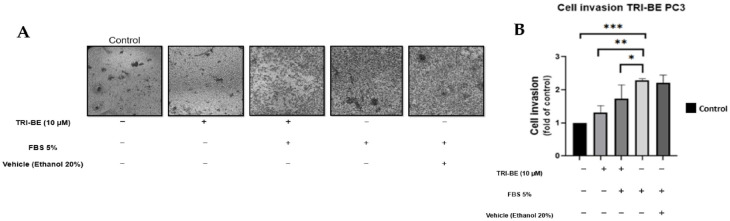
TRI-BE decreases the invasive capacity of PC3 cells. (**A**) A 24 h time course. Treatments are plotted from left to right: (control) DMEM; (TRI-BE) 10 µM TRI-BE + DMEM; (TRI-BE/FBS) 10 µM TRI-BE + DMEM + FBS 5%; (FBS) DMEM + FBS 5%; (Vehicle) DMEM + ethanol 20%. (**B**) The graph represents the means of each treatment ± SD of at least three independent experiments. Asterisks indicate statistical significance; * *p* < 0.05, ** *p* < 0.01, *** *p* < 0.001, assessed using one-way ANOVA with Dunnett’s test.

**Figure 6 molecules-28-04524-f006:**
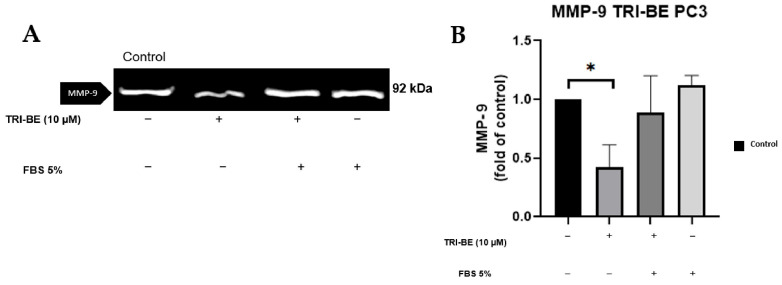
Effect of TRI-BE on MMP-9 secretion. (**A**) A 24 h time course. Treatments are plotted from left to right: (control) DMEM; (TRI-BE) 10 µM TRI-BE + DMEM; (TRI-BE/FBS) 10 µM TRI-BE + DMEM + FBS 5%; (FBS) DMEM + FBS 5%. (**B**) The graph represents the means of each treatment ± SD of at least three independent experiments. Asterisks indicate statistical significance; * *p* < 0.05, as assessed using one-way ANOVA with Dunnett’s test.

**Figure 7 molecules-28-04524-f007:**
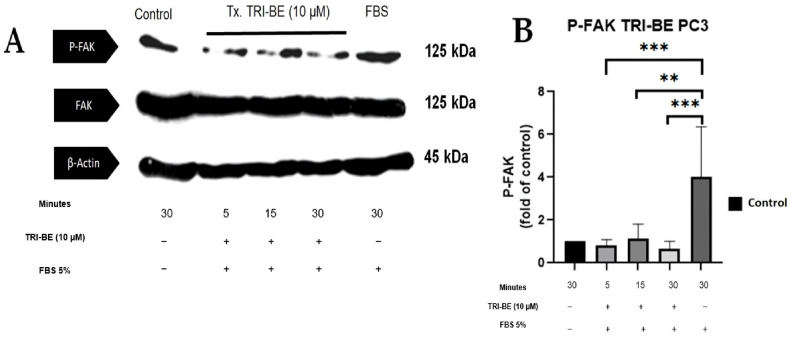
TRI-BE induces a decrease in phosphorylated FAK (p-FAK) levels. (**A**) Time courses of 5, 15, and 30 min. Treatments are plotted from left to right: (control) DMEM; TRI-BE (5 min) 10 µM TRI-BE + DMEM + FBS 5%; TRI-BE (15 min) 10 TRI-BE + DMEM + FBS 5% (FBS); TRI-BE (30 Min) 10 µM TRI-BE + DMEM + FBS 5%; (FBS) DMEM + FBS 5%. (**B**) The graph represents the means of each treatment ± S.D. of at least three independent experiments. Asterisks indicate statistical significance; ** *p* < 0.01, *** *p* < 0.001, as assessed using one-way ANOVA with Dunnett’s test.

**Figure 8 molecules-28-04524-f008:**
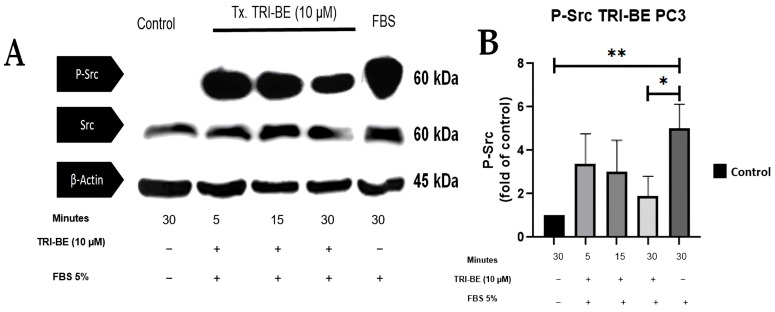
TRI-BE induces a decrease in p-Src levels in PC3 cells. (**A**) Temporary courses of 5, 15, and 30 min. Treatments are plotted from left to right: (CTRL) DMEM; TRI-BE (5 min) 10 µM TRI-BE + DMEM + FBS 5%; TRI-BE (15 min) 10 µM TRI-BE + DMEM + FBS 5%; TRI-BE (30 min) 10 TRI-BE + DMEM + FBS 5%; (FBS) DMEM + FBS 5%. (**B**) The graph represents the means of each treatment ± SD of at least three independent experiments. Tx = treatment. Asterisks indicate statistical significance; * *p* < 0.05, ** *p* < 0.01, as assessed using one-way ANOVA with Dunnett’s test.

**Figure 9 molecules-28-04524-f009:**
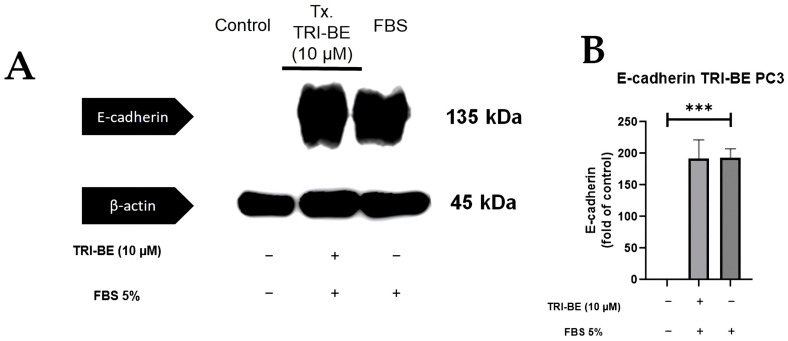
Effect of TRI-BE on E-cadherin expression levels. (**A**) A 24 h time course. Treatments are plotted from left to right: (control) DMEM; TRI-BE 10 µM TRI-BE + DMEM + FBS 5%; (FBS) DMEM + FBS 5%. (**B**) The graph represents the means of each treatment ± SD of at least three independent experiments. Asterisks indicate statistical significance; *** *p* < 0.001, as assessed using one-way ANOVA with Dunnett’s test.

**Figure 10 molecules-28-04524-f010:**
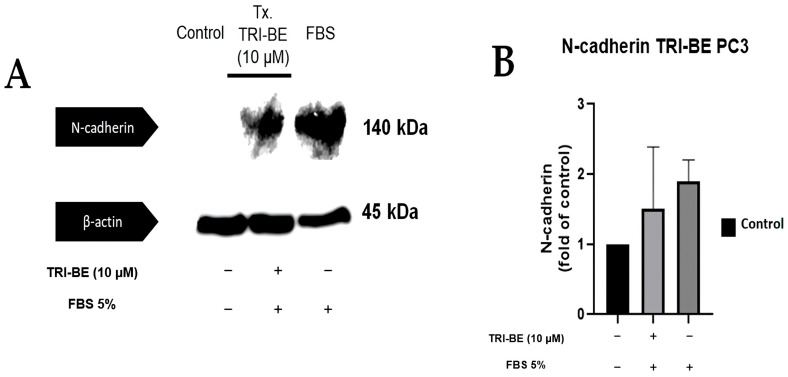
Effect of TRI-BE on N-cadherin expression levels. (**A**) A 24 h time course. Treatments are plotted from left to right: (control) DMEM; TRI-BE + DMEM + FBS 5%; (FBS) DMEM + FBS 5%. (**B**) The graph represents the means of each treatment ± SD of at least three independent experiments evaluated using one-way ANOVA with Dunnett’s test.

**Figure 11 molecules-28-04524-f011:**
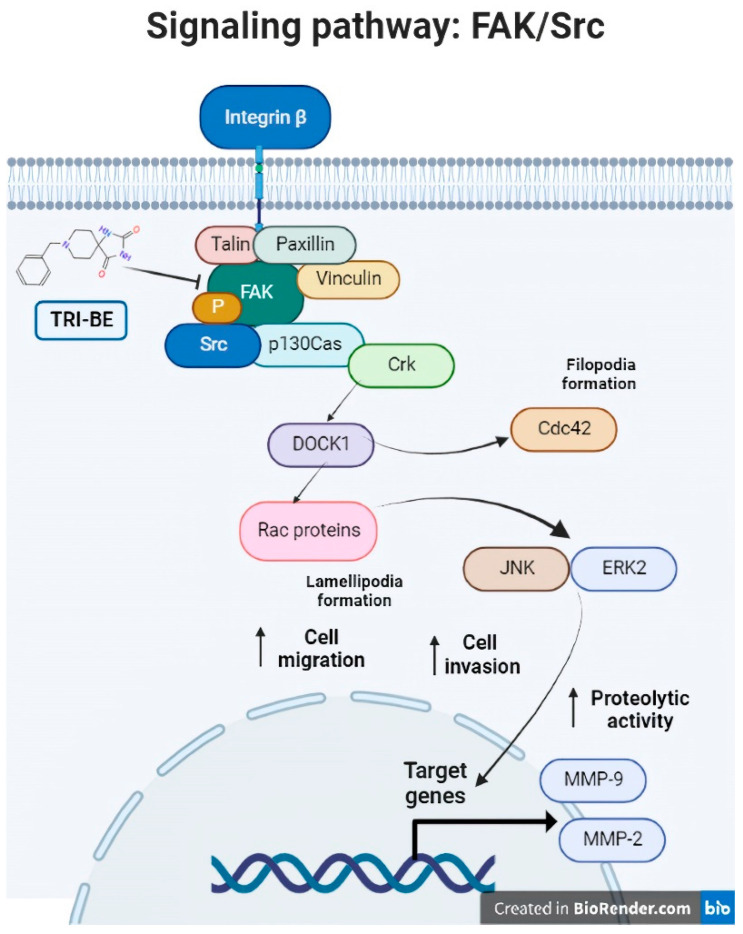
The FAK/Src signaling pathway. Association of the FAK/Src signaling pathway with various mechanisms involved in tumor progression (angiogenesis and cell migration and invasion). [from the figure filopodia formation].

**Table 1 molecules-28-04524-t001:** The highest-affinity E-score values obtained from molecular docking of TRI-BE and FAK.

Analysis	E-Score (kcal/mol)
FAK TRI-BE (A)	−7.7878
FAK TRI-BE (B)	−7.7637
FAK TRI-BE (C)	−7.6901

**Table 2 molecules-28-04524-t002:** The highest-affinity E-score values obtained from molecular docking of TRI-BE and Src.

Analysis	E-Score (kcal/mol)
Src TRI-BE (A)	−8.2888
Src TRI-BE (B)	−7.9860
Src TRI-BE (C)	−7.8069

## Data Availability

The data obtained throughout the experiments can be provided by O. G-H. upon reasonable request.

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
