# Peer review of "A Triazaspirane Derivative Inhibits Migration and Invasion in PC3 Prostate Cancer Cells"

_molecules, 2023, doi:10.3390/molecules28114524_

Round 1

Reviewer 1 Report

The manuscript under review describes the potential activity of triazaspirane conformers as inhibitors of PC3 prostatic tumor cells. The study is based on a molecular docking study  and some biological assays to demonstrate the molecule effect on tumor cells migration and invasion.

The study is well designed, results are very interesting and manuscript well written.

I have few comments:

- Why triazispiranes? Authors should briefly explain why they have selectet these structures.

- A brief comment on the different  conformers and how their conformation affects  the bioactivity should be added.

- A list of abbreviation should be added

- Fig. 1 shoudl be improved.

Attached you find other suggestions which I have directly added to the text.

Author Response

Reviewer 1

The manuscript under review describes the potential activity of triazaspirane conformers as inhibitors of PC3 prostatic tumor cells. The study is based on a molecular docking study and some biological assays to demonstrate the molecule effect on tumor cells migration and invasion.

The study is well designed, results are very interesting and manuscript well written.

I have few comments:

Why triazispiranes? Authors should briefly explain why they have selectet these structures.

Response:  According to the suggestion, the following information was added in the Introduction section, page 2.

Azaspiran-type molecules are compounds that have been associated with various beneficial effects, including their immunoregulatory capacity in murine models with rheumatoid arthritis, decreasing the inflammatory response resulting in an improvement in the preservation of bone integrity (16, 17).

On the other hand, antitumor effects have been described in azaspiran-type molecules. In myeloma cells, N'N-diethyl-8,8-dipropyl-2-azaspiro[4,5]decane-2-propanamine (Atiprimod) exhibits anti-proliferative and anti-angiogenic properties. Atiprimod modulates these processes through activation of pro-apoptotic proteins such as caspase-3 and -8, inhibition of Akt and STAT3 phosphorylation, and decreased production of vascular endothelial growth factor (VEGF) and interleukin. 6 (IL-6) (18,19); coupled with a decrease in the protein expression levels of JAK2 and p-JAK2, negatively regulating the JAK-STAT pathway (20,21). In line with this notion, the anti-proliferative effect of a derivative of the type 1-oxa-3-azaspiro[5.5]undecane has been observed through the modulation of the JAK2/STAT3 pathway in hepatocellular carcinoma and breast cancer cells, regardless of estrogen receptor expression status (14,22).

A brief comment on the different  conformers and how their conformation affects  the bioactivity should be added.

Response: According to the suggestion, the following information was added in Results section, page 6.

The three conformers with the lowest binding energy were selected, since it is indicative of a higher degree of affinity to bind to the amino acids of a specific region of the FAK and Src proteins. These values are influenced by the interactions present in the coupling, such is the case of van der Waals, electrostatic, hydrogen bond and hydrophobic interactions  (Ref. Molecular Docking: A powerful approach for structure-based drug discovery). According to what is described in the interactions of Figure 1, it can be established, at least theoretically, that the coupling of TRI-BE with FAK has a great affinity for the region of the kinase domain, an important domain since it generates the phosphorylation of target proteins downstream of FAK that lead to adhesion, spreading, migration and metastasis. Moreover, catalytic site binding proteins such as FIP200 have been shown to inhibit FAK activity.

On the other hand, in coupling with the Src protein in Figure 2, the selected TRI-BE conformers show higher affinity for the SH2 and SH3 domains, important domains for interaction with proteins. Particularly, the binding of TRI-BE to the SH2 domains in Src would block the recognition and interaction with the phosphorylated Y397 residue of FAK, suppressing the activation of the FAK/Src pathway (Ref.The dual kinase complex FAK-Src as a promising therapeutic target in cancer) accompanied by a lower migratory and invasive capacity of cancer cells. This mechanism would explain the inhibitory effect of TRI-BE on the migration and invasion of PC3 cells.

A list of abbreviation should be added

Response: According to the suggestion, the following information was added in page 1.

List of Abbreviations : TRI-BE, 8-Benzyl-1,3,8-triazaspiro[4.5]decane-2,4-dione; FAK, focal adhesion kinase; CIMO, 2-(1-(4-(2-cyanophenyl)1-benzyl-1H-indol-3-yl)-5-(4-methoxyphenyl)-1-oxa-3-azaspiro-(5,5)undecane; PVDF, Polyvinylidene Difluoride; SDS, Sodium Dodecyl Sulfate; SDS-PAGE, sodium dodecyl sulfate polyacrylamide gel electrophoresis; FBS, fetal bovine serum; PBS, phosphate-buffered saline; MOE, Molecular Operating Environment, PDB, Protein Data Bank; MMP-2, matrix metalloproteinase-2; MMP-9, matrix metalloproteinase-9, ECM, extracellular matrix.

Fig. 1 shoudl be improved.

According to the suggestion, Fig 1 was modified and improved. Fig 1A includes the molecular structure of TRI-BE.

Attached you find other suggestions which I have directly added to the text.

Based on the recommendations that you kindly indicated to us, the modifications were made directly to the manuscript. Briefly removed suggested text sections. In addition, the initial section on Materials and Methods was relocated to the Introduction section (page 2).

Reviewer 2 Report

What I miss in the introduction is information explaining why this particular molecule is being studied. The reference to one publication about a similar active compound on a different cell type is insufficient.

In my opinion, the structural formula of the molecule under investigation should be given, as well as the formulae of other derivatives , for example those referred to by symbols in the discussion. The absence of these structures does not allow conclusions to be drawn on the structure-activity relationship for this type of compounds. Which elements of the structure must be retained for a compound to be active and which can be changed to improve activity?

In the experimental part, I did not find information on where the compound was taken from, whether bought or synthesised. How it was prepared for testing - solvent and concentration.

The test concentration at which the effect of the compound was observed is quite high. Was cytotoxicity to PC3 prostate cancer cells determined?

What I find missing from the discussion is the element I point out in the introduction - the activity of the compound in relation to its structure and the structures of similar derivatives. 

In the table of references, the double numbering and the doi numbers must be removed.

Author Response

Reviewer 2

What I miss in the introduction is information explaining why this particular molecule is being studied. The reference to one publication about a similar active compound on a different cell type is insufficient.

Response:  According to the suggestion, the following information was added in the Introduction section, page 2

Azaspiran-type molecules are compounds that have been associated with various beneficial effects, including their immunoregulatory capacity in murine models with rheumatoid arthritis, decreasing the inflammatory response resulting in an improvement in the preservation of bone integrity (16, 17).

On the other hand, antitumor effects have been described in azaspiran-type molecules. In myeloma cells, N'N-diethyl-8,8-dipropyl-2-azaspiro[4,5]decane-2-propanamine (Atiprimod) exhibits anti-proliferative and anti-angiogenic properties. Atiprimod modulates these processes through activation of pro-apoptotic proteins such as caspase-3 and -8, inhibition of Akt and STAT3 phosphorylation, and decreased production of vascular endothelial growth factor (VEGF) and interleukin. 6 (IL-6) (18,19); coupled with a decrease in the protein expression levels of JAK2 and p-JAK2, negatively regulating the JAK-STAT pathway (20,21). In line with this notion, the anti-proliferative effect of a derivative of the type 1-oxa-3-azaspiro[5.5]undecane has been observed through the modulation of the JAK2/STAT3 pathway in hepatocellular carcinoma and breast cancer cells, regardless of estrogen receptor expression status (14,22).

In my opinion, the structural formula of the molecule under investigation should be given, as well as the formulae of other derivatives , for example those referred to by symbols in the discussion . The absence of these structures does not allow conclusions to be drawn on the structure-activity relationship for this type of compounds. Which elements of the structure must be retained for a compound to be active and which can be changed to improve activity?

Response: According to the suggestion, Fig 1 was modified and improved. Fig 1A includes the molecular structure of TRI-BE. Additionally, in  Fig. supplementary 1, we indicate the structures of other azaspiran derivatives such as Atriprimod and CIMO.

In addition, the following information was added in the Discussion section, page 11.

Spiro-type compounds are those that have a structure formed by two orthogonally joined rings, these compounds usually have different heterocyclic components, managing to acquire a certain similarity with some macromolecules such as proteins, allowing them to obtain the ability to interact with proteins in complex biological systems (Ref. Effects of Spiro-bisheterocycles on Proliferation and Apoptosis in Human Breast Cancer Cell Lines). 

In multiple cancer models, azaspirans and their derivatives have been tested as inhibitors of some tyrosine kinase proteins, related to tumor progression, such as atiprimod (JAK2/JAK3 inhibitor) and lestaurtinib (JAK2 inhibitor), which present different tumor inhibition mechanisms (Ref. Development of a Novel Azaspirane That Targets the Janus Kinase-Signal Transducer and Activator of Transcription (STAT) Pathway in Hepatocellular Carcinoma in Vitro and in Vivo) (Ref. An azaspirane derivative suppresses growth and induces apoptosis of ER-positive and ER-negative breast cancer cells through the modulation of JAK2/STAT3 signaling pathway).

In particular, the cyclic structure of TRI-BE is the spirohydantoin type, where hydantoin has been used for a wide variety of pathologies, notable for its anticonvulsant, antiinflammatory, and antitumor properties, among others (Ref. An insight into the structure of 5 -spiro aromatic derivatives of imidazolidine-2,4-dione, a new group of very potent inhibitors of tumor multidrug resistance in T-lymphoma cells).

Figure supplementary 1. A) 8-Benzyl-1,3,8-triazaspiro[4.5]decane-2,4-dione (TRI-BE). B) N-N-diethyl-8,8-dipropyl-2-azaspiro [4.5] decane-2-propanamine (Atiprimod). C) 2-(1-(4-(2-cyanophenyl)1-benzyl-1H-indol-3-yl)-5- (4-methoxy-phenyl)-1-oxa-3-azaspiro(5,5)undecane (CIMO).

In the experimental part, I did not find information on where the compound was taken from, whether bought or synthesised. How it was prepared for testing - solvent and concentration.

Response: Based on your kind suggestion, we have added the following information in the Materials and Methods section, page 14.

Compound preparation

The TRI-BE molecule was synthesized at the Graduate and Research Center of the Technological Institute of Tijuana, Baja California, according to the methodology described by Rivero et al.[ 52]. Once the compound was obtained in its pure state, it was diluted in a 20% (v/v) ethanol solution (vehicle) to a concentration of 1 uM (stock solution). Subsequently, the stock solution was used to stimulate the PC3 cell cultures at the concentrations indicated in the experiments.

The test concentration at which the effect of the compound was observed is quite high. Was cytotoxicity to PC3 prostate cancer cells determined?

Response: Multiple studies have evaluated the cytotoxic effect of molecules derived from azaspirans. In the case of atiprimod, through an MTT assay, it was observed that this molecule was cytotoxic at concentrations greater than 3 µM, in GH3 murine pituitary adenoma cells (Atiprimod induce apoptosis in pituitary adenoma: Endoplasmic reticulum stress and autophagy pathways) (2019). On the other hand, Zeslawka and collaborators analyzed a series of spirohydantoin-type compounds in T-lymphoma cells, observing IC50 values in a range of 10 µM to 100 µM in all the compounds analyzed. (An insight into the structure of 5-spiro aromatic derivatives of imidazolidine-2,4-dione, a new group of very potent inhibitors of tumor multidrug resistance in T-lymphoma cells) (2021). In line with this nation, the effect of CIMO on LO2 cells was evaluated. The data showed that exposure of LO2 cells with 100 µM CIMO for 72 h did not generate cytotoxic effects Development of a Novel Azaspirane That Targets the Janus Kinase-Signal Transducer and Activator of Transcription (STAT) Pathway in Hepatocellular Carcinoma in Vitro and in Vivo) (2014). Additionally, Ramdanie et al. evaluated the effect of a series of heterocyclic spiro-type compounds on cell proliferation using MCF-7 and MDA-MB-231 breast cancer cell lines. Some of the selected compounds showed an IC50 of 42.3 and 66 µM in MCF-7 cells, while IC50 of 67.9 and 97 µM were observed for MDA-MB-231 cells (Effects of Spiro-bisheterocycles on Proliferation and Apoptosis in Human Breast Cancer Cell Lines) (2016). Furthermore, leukemia cells were exposed for 72 h to molecules derived from spirohydantoin at different concentrations (10-250 µM). Their data indicate that these molecules induce cytotoxicity from 50 µM (Kavitha et al), Synthesis and in vitro cytotoxic evaluation of novel diazaspiro bicyclo hydantoin derivatives in human leukemia cells: a SAR study). Taken together, these data indicate that spirohydantoin-type molecules present cytotoxicity at higher concentrations than those used in our experiments.

Here, we show an experiment of migration assays with original photos in which PC3 cells are exposed to a concentration of 50 µM, no cell damage or suspended cells are observed  (Fig. 4B). This strongly suggests that high concentrations of TRI-BE do not promote cytotoxicity in PC3 cells, but inhibit migration at a concentration of 10 uM.

Figure 4B. TRI-BE inhibits PC3 cell migration. A) A 24-h time course. Treatments are plotted from left to right: (Control) DMEM; (FBS) DMEM + FBS 5%; (Vehicle) DMEM + FBS 5% + Ethanol 20%; (TRI-BE 10 µM) 10 µM TRI-BE + DMEM + FBS 5%; (TRI-BE 50 µM) 50 µM TRI-BE + DMEM + FBS 5%.

Your recommendation was considered, and we incorporated a new figure 4A into the manuscript and we add information in Results and the Materials and methods section. In this figure, we indicate that TRI-BE has no effect on the viability of PC3 cells. 

What I find missing from the discussion is the element I point out in the introduction - the activity of the compound in relation to its structure and the structures of similar derivatives.

Response: The following information was added in the Discussion section, page 11.

Spiro-type compounds are those that have a structure formed by two orthogonally joined rings, these compounds usually have different heterocyclic components, managing to acquire a certain similarity with some macromolecules such as proteins, allowing them to obtain the ability to interact with proteins in complex biological systems (Ref. Effects of Spiro-bisheterocycles on Proliferation and Apoptosis in Human Breast Cancer Cell Lines). 

In multiple cancer models, azaspirans and their derivatives have been tested as inhibitors of some tyrosine kinase proteins, related to tumor progression, such as atiprimod (JAK2/JAK3 inhibitor) and lestaurtinib (JAK2 inhibitor), which present different tumor inhibition mechanisms (Ref. Development of a Novel Azaspirane That Targets the Janus Kinase-Signal Transducer and Activator of Transcription (STAT) Pathway in Hepatocellular Carcinoma in Vitro and in Vivo) (Ref. An azaspirane derivative suppresses growth and induces apoptosis of ER-positive and ER-negative breast cancer cells through the modulation of JAK2/STAT3 signaling pathway).

In particular, the cyclic structure of TRI-BE is the spirohydantoin type, where hydantoin has been used for a wide variety of pathologies, notable for its anticonvulsant, antiinflammatory, and antitumor properties, among others (Ref. An insight into the structure of 5 -spiro aromatic derivatives of imidazolidine-2,4-dione, a new group of very potent inhibitors of tumor multidrug resistance in T-lymphoma cells).

In the table of references, the   and the doi numbers must be removed.

We appreciate your suggestions. The style of the reference list has been improved.
